# Feasibility of the International Wealth Index and the Gapminder tool as instruments to assess household income and estimate catastrophic expenditure: A prospective patient-level cohort study in India

**CROCODILE study group**\*¶

¶ Membership of the CROCODILE study group is provided in S1 Appendix.
\* jfs945@bham.ac.uk

**Data Availability Statement:** Data is only available to the participating researchers, as per the

## Abstract

### Background

Patient income assessment is required to assess healthcare catastrophic expenditure (Sustainable Development Goal) but self-reported income has several biases. This study aimed to assess the feasibility of the International Wealth Index (IWI) and the Gapminder tool as indirect instruments to assess household income.

### Methods

Prospective cohort study of colorectal cancer patients in five tertiary care hospitals in India (Dec 2020-August 2021). Patient self-reported household income was compared to income estimated from the IWI (twelve questions about household goods) and the Gapminder tool (five pictures of household assets). Agreement between instruments was explored with Bland-Altman methods. Cancer care expenditure from the same cohort was used to illustrate the impact of these tools in catastrophic expenditure rates.

### Results

From the 226 patients included, 99.5% completed the IWI and the Gapminder tool. Overall, self-reported incomes were lower than the estimated from the IWI and Gapminder tools (median incomes: 17350₹ for self-reported, 37491₹ for IWI and 51520₹ for Gapminder). The IWI showed better agreement with the self-reported income than the Gapminder tool. For both instruments, the agreement was better for low income earning households. Illustrative catastrophic expenditure rates range from 71% to 56% to 43% when using self-reported, IWI and Gapminder incomes respectively.

regulations of the Indian Council of Medical Research, as data sharing agreements need to be in place for data sharing. Data are available from the NIHR Research Unit on Global Surgery Institutional Data Access / Ethics Committee (contact via jfs945@bham.ac.uk) for researchers who meet the criteria for access to confidential data.

**Funding:** This work was funded by a National Institute for Health Research (NIHR) Global Health Research Unit Grant (NIHR 16.136.79) using UK aid from the UK Government to support global health research. The funders had no role in study design, data collection, analysis and interpretation, or writing of this report. The views expressed are those of the authors and not necessarily those of the National Health Service, the NIHR, or the UK Department of Health and Social Care.

**Competing interests:** The authors have declared that no competing interests exist.

## Discussion

It is feasible to use the IWI and the Gapminder tools to estimate household income although they might overestimate income, with an impact on catastrophic expenditure rates. Further refinement of these tools could enable global monitoring and modelling of catastrophic expenditure from real-world data, at low burden for patients.

## Introduction

Worldwide, more than 900 million people are experiencing healthcare related catastrophic expenditure and households with non-communicable diseases, particularly cancer are at high risk [1, 2]. Cancer patients are a particularly vulnerable group and a recent meta-analysis identified that 43% are suffering financial catastrophe due to cancer care [3]. The reduction of catastrophic and impoverishing health expenditure has been prioritised in the Sustainable Development Goals of the World Health Organisation (SCG indicator 3.8.2), to improve access to healthcare and promote and social financial protection for families [4]. Patients are considered to be undergoing catastrophic expenditure if their out-of-pocket payments for healthcare are higher than a proportion of their household income (10 to 25% when using total household income or 40% when using non-subsistence income). The assessment of patient household income is a necessary step to measure catastrophic expenditure rates, monitoring the progress of global health aims and identifying vulnerable populations.

Self-reported income is widely used to assess household income (e.g. household surveys) but is exposed to several biases and limitations. Data from previous studies showed that patients do feel reluctant to answer questions about income and that this can lead to high rates of non-response [5]. Individual characteristics can influence the likelihood of accurate income reporting and surveyed people often have difficulty in interpreting formulated questions about income [6, 7]. Additionally, social desirability bias is commonly present in household surveys responses, introducing over or underestimations depending on the socio-economic and cultural setting, as well as individual circumstances at the point of data collection [8, 9]. These limitations of self-reported income can have an impact on population-level estimations (e.g. poverty lines) and policy decision making at national and global level [10, 11].

To the best of our knowledge, direct income reporting is the standard method to assess income and there are no alternatives. A search on PubMed and Google Scholar was performed identifying one study where an asset index was assessed as a tool to predict income quintiles with low performance in African populations and an exploratory analysis from data from Mexico showing that concluding that assets might better reflect wealth as they fluctuate less over time [12, 13]. A systematic review exploring the relationship between wealth indexes and socio-economic status demonstrated that they are likely to be distinct measures [14]. No previous studies were found, in which asset or house characteristics were used to estimate household income figures for catastrophic expenditure calculations. This study aimed to assess the feasibility of the International Wealth Index (IWI) and the Gapminder tool as indirect measures of household income, in a cohort of patients with colorectal cancer in India.

## Material and methods

### Study design and setting

This is a prospective multicentric cohort study conducted in tertiary care hospitals in India, including consecutive patients with a new treatment decision for colorectal cancer from

December 2020 to August 2021. The overall goal of this cohort study was to determine catastrophic expenditure rates among colorectal cancer patients in India. In order to assess catastrophic expenditure, both out-of-pocket payments for cancer and patient household income assessment were collected. This is a pre-planned analysis comparing three different instruments of assessing patient household income, to assess the feasibility of their use.

The study protocol has been published and the study is registered on ClinicalTrials.gov (NCT04517032) and with the Central Trials Registry of India (CTRI/2020/09/027896) [15]. Indian Council of Medical Research, University of Birmingham and hospital level ethical approvals were obtained for the study. Written informed consent was obtained from all patients, as per local ethical requirements. The hospitals started data collection as soon as their ethical approval was obtained, recruiting a maximum of seventy patients per hospital until the total sample size was achieved. The sample size was defined at the time of protocol design, assuming an anticipated proportion of cancer patients suffering catastrophic expenditure around 45% [13, 14]. For a prespecified absolute precision of 10% and 5% (error margins recommended by the United Nations for household surveys), a respective sample size of 95–380 patients would be required at a confidence interval of 95% [15]. A pragmatic trade-off was decided according to usual recruitment numbers at colorectal cancer multidisciplinary teams in the included hospitals, allowing for a feasible study length and clinical relevance.

## Patient inclusion and data collection

All consecutive patients with a new treatment decision for colorectal cancer were identified from multidisciplinary team meetings or outpatient clinics, depending on local pathways. Recruitment and baseline data collection were conducted in the first hospital visit after patient identification, during which patients completed a self-reported income assessment as well as the International Wealth Index and the Gapminder tool.

Further data variables were collected: age, sex, education level (primary, secondary, graduate, post-graduate, did not attend school), job skill level (categorised according to the International Standard Classification of Occupations [16] as high, medium, low or not specified), distance from the patients' house to the hospital and number of people living in the household (number of adults and children sharing the same house). Tumour and treatment plan details were also collected at the point of patient inclusion: tumour location (colon or rectum), tumour stage (local or advanced, as per AJCC 8[th] edition Tumour Node Metastasis classification) and treatment intent (curative or palliative). Follow-up data was collected at 6 weeks, 3 months and 6 months after the baseline assessment, including details of the treatment course and payments made by patients for cancer care (direct medical, non-medical and indirect payments are fully described in S1 Table in S1 Appendix).

## Income assessment instruments

**Self-reported income.** Patients reported the total amount of money earned monthly by all the household members who work and produce income, in Indian Rupees (INR, ₹). This could include a regular monthly income in the form of a salary or an average monthly income as a result of a variable daily wage.

**International Wealth Index.** The International Wealth Index (IWI) is a twelve-point questionnaire where patients reported whether they owned a number of consumer durables (television, refrigerator, phone, car, bicycle, cheap utensils e.g. chair and expensive utensils e.g. air conditioner) and described their household characteristics (floor material, toilet facility, number of rooms, access to electricity and water source) [17]. Based on the responses to these questions, a final score is calculated, ranging from 0 to 100 where a low score relates to lower

wealth levels and a high score relates to a wealthier household. The full score is available in the
S1 Appendix: Details of the income assessment instruments.

The final IWI score was translated into an income figure using a previously available for-
mula that explains the relationship between income and the IWI score in 2015 United States
Dollars (USD) [18]. This formula [18] was obtained by computing the IWI in multiple popula-
tion based household surveys (including the India Human Development Survey but also popu-
lation datasets from other low-middle income countries), on the basis of the available
information about households and assets. The formula estimates how an increase in the IWI
score is associated with an expected average increase in household income.

$$IWI\ income\ (per\ day,\ in\ USD) = 1.489056 * exp(0.02918)$$

The formula above correlates the IWI with an income figure in 2015 United States Dollars
(USD). In order to obtain a figure that is comparable to the income reported by patients in our
study in 2021 (Indian Rupees), the IWI income in USD was multiplied by purchase power par-
ities (PPP) for USD-INR exchange and by the consumer price index to account for inflation
between 2015 and 2021 [19].

**Gapminder tool.** The Gapminder Foundation has created the Dollar Street project where
a pool of pictures of household and personal goods are available and matched to an income
reported by the household members to the Gapminder team [20]. The methods used by the
Gapminder team to assess household income and match them to the pictures were varied,
including: reported income, reported consumption, owned assets, minimum and average
wages for people's occupations and benefits available through social welfare. The patients were
shown five categories of pictures from Indian households (all the available at the time of study
design): houses, kitchens, floors, toilets and beds. The patient was invited to select the house,
kitchen, floor, toilet and bed that looked more similar to the one their own, to which an
income was then matched (more details are available in the S1 Appendix: Details of the income
assessment instruments). The mean average of the income matched to the selected pictures
was obtained and converted from USD to INR using purchase power parities, as follows:

$$Gapminder\ income = \frac{(Income\ matched\ to\ house + kitchen + floor + toilet + bed)}{5} * PPP$$

**Statistical analysis and data handling.** Continuous variables were described with stan-
dard summary metrics: mean and standard deviation when normally distributed and median
and inter-quartile range (IQR) when skewedly distributed. Categorical variables were
described using frequency tables. A Bland-Altman analysis was performed to describe the
agreement between (1) self-reported and IWI income and (2) self-reported and Gapminder
income. In the Bland-Altman plots, the difference between reported and estimated incomes
was plotted against the mean average of those the same incomes [21]. A standard Bland-Alt-
man analysis with agreement limits based on 95% confidence intervals for the observations
was not possible, provided that the income data did not comply with normality assumptions.
We opted for the description of the proportion of patients for whom the differences between
reported and estimated incomes were within a particular range (methodology suggested by
Bland and Altman for non-parametric data [22]. This allows an understanding of the range of
differences between reported and estimated incomes, as well as the description of any patterns
in the Bland-Altman plots. Missing data was described for all variables in the tables and plots.

**Illustration of the impact of income assessments in catastrophic expenditure rates.**
Cancer care related expenditure was collected from the same cohort of patients, at 6 weeks

after a new decision for treatment by the clinical team in charge of the patient (costs were assessed at other timepoints and will be published separately). This was used to illustrate the impact of the different income assessment tools in catastrophic expenditure rates. Catastrophic expenditure was defined as out-of-pocket payments for cancer care being 25% of total household income for the purpose of this analysis, as per the most updated WHO definition of the Sustainable Development Goals [4]. Full description of the out-of-pocket payments included in this illustration are available in S1 Table in S1 Appendix.

## Results

### Patient characteristics

In total, 226 patients were included from five tertiary care hospitals in India. The median age was 52 years and 63.3% (143/226) of the patients were men. The household size was of one to three people in 26.5% of the patients (60/226) and four to six people in 52.2% of the patients (118/226). Overall, 104 of all patients did not have a job (being retired or housewives), with the remaining patients having a similar distribution in terms of professional skill levels (low 13.3%, medium 17.7% and high 17.7%). Regarding the education levels, the majority of the patients completed primary (22,6%), secondary (36.7%) or graduate degrees (27.9%), with a minority not having attended school (4.4%). Regarding clinical features, 118 patients had rectal cancer and 108 had colon cancer. Most of the patients had a treatment plan with a curative intent (83.1% [187/226]) and 16.9% received palliative treatment only. For full patient characteristics see Table 1.

### Self-reported, IWI and Gapminder income distribution

From the 226 patients included, 99.5% completed all the three instruments of income assessment. Only one patient did not complete the IWI and another patient did to respond to the Gapminder tool. Overall, the median reported income (17350 ₹, (IQR 10000–40000)) was lower than median income assessed using the IWI (37491 ₹, (IQR 25421–61322)) and from the Gapminder tool (51520 ₹, (IQR 22120–91128)). The distribution and metrics of the self-reported IWI and Gapminder incomes are shown in Fig 1.

The distribution of the self-reported incomes was more skewed to the left (lower incomes), meaning that more patients reported a lower income directly, when compared to the incomes estimated with the IWI and the Gapminder tool (Fig 2). The distribution of IWI income figures is more homogeneous, and the figures are contained within a narrower range, when compared to the Gapminder or the self-reported income where more outliers exist. The Gapminder income follows a similar distribution when compared to the reported income but with a flattened density curve, where the results are not as skewed towards low incomes as they are for self-reported income.

S1(A) and S1(B) Fig in S1 Appendix shows scatter plots exploring the association between the self-reported income and estimated incomes using the IWI and the Gapminder tool respectively. Both estimated incomes show a positive non-linear association with self-reported incomes. The association is strong for low reported incomes, becoming weaker as the reported income increases.

### Agreement between self-reported, IWI and Gapminder incomes

The differences between reported and estimated incomes show lower dispersion for the IWI, compared to the Gapminder income (Fig 3A). The proportion of patients for whom the estimated income is within 50000 INR of the self-reported income is 92.8% for the IWI and 70.2%

**Table 1. Demographic and tumour characteristics of the included patients.**

| Variable | Category | n(%) |
|---|---|---|
| Age | 18–30 years | 18 (8.0) |
| | 31–50 years | 83 (36.9) |
| | 51–70 years | 96 (42.7) |
| | >70 years | 28 (12.4) |
| | (Missing) | 1 |
| Sex | Male | 143 (63.3) |
| | Female | 83 (36.7) |
| Education level | Primary | 51 (22.6) |
| | Secondary | 83 (36.7) |
| | Graduate | 63 (27.9) |
| | Post-graduate | 19 (8.4) |
| | Did not attend school | 10 (4.4) |
| Job skill level | Low | 30 (13.3) |
| | Medium | 40 (17.7) |
| | High | 40 (17.7) |
| | Not specified—Housewife | 65 (28.8) |
| | Not specified—Retired | 39 (17.3) |
| | Not specified—Other | 12 (5.3) |
| Household size | 1–3 people | 60 (26.5) |
| | 4–6 people | 118 (52.2) |
| | 7–9 people | 35 (15.5) |
| | 10+ people | 13 (5.8) |
| Distance from home | 0–100 km | 77 (34.1) |
| | 100–500 km | 73 (32.3) |
| | 500-1000km | 10 (4.4) |
| | >1000 km | 66 (29.2) |
| Hospital type | Government | 80 (35.4) |
| | Charity / Private | 146 (64.6) |
| Cancer location | Colon | 108 (47.8) |
| | Rectum | 118 (52.2) |
| Cancer stage | Local | 31 (13.7) |
| | Advanced | 195 (86.3) |
| Treatment intent | Curative | 187 (83.1) |
| | Palliative | 38 (16.9) |
| | (Missing) | 1 |

Data reported with frequency and proportions as n (%).

for the Gapminder tool. Similar trends are observed for differences within lower limits (25000 INR) and higher limits (100000 INR), reinforcing better agreement for the IWI (see Fig 3B).

In the Bland-Altman plots describing agreement for both the IWI and the Gapminder tool (Fig 3A), there is a funnel shaped distribution of the differences between self-reported and estimated incomes. The differences are closer to zero for low income earning households and farther from zero as the household income increases, demonstrating a worse degree of agreement for high income earning households. Standard 95% confidence interval for agreement can't be drawn for this data as the differences between income assessments didn't follow a normal distribution (Q-Q plots shown in S2 Fig in S1 Appendix).

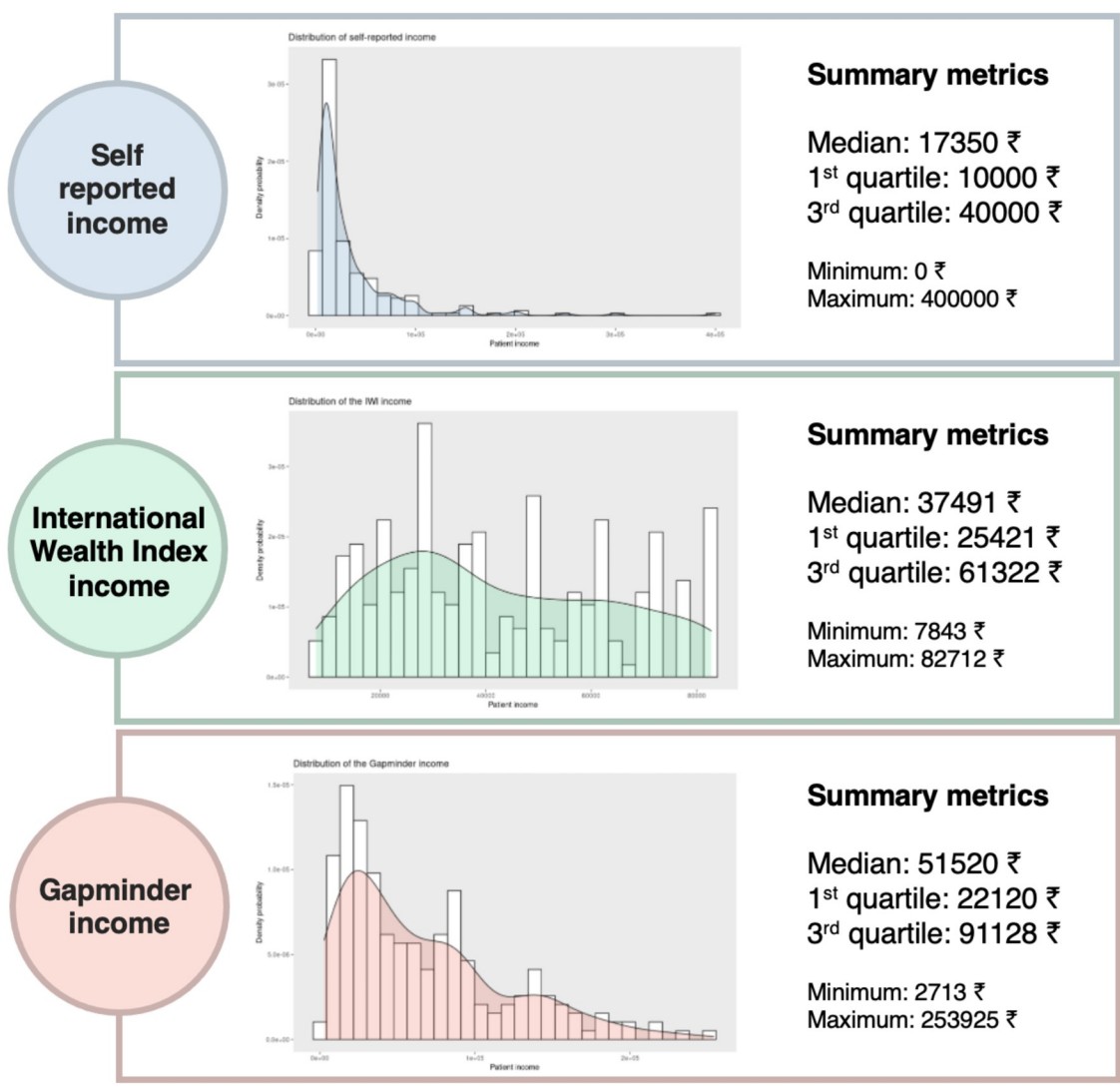

**Fig 1. Distribution and summary metrics of patient household monthly income, assessed by self-reporting, the International Wealth Index and the Gapminder tool.** The graphs display histograms of the self-reported, IWI and Gapminder incomes, with overlapped Kernel density plots showing the overall distribution of income figures in the three types of assessment. Summary metrics of the three income assessments are provided to allow comparison. All incomes are reported in Indian Rupees (₹). *Full description of the income methods assessment available in Methods.*

### Illustration of catastrophic expenditure rates

Within six weeks of having received a new decision for treatment, this cohort of patients had a median expenditure of 202291INR for cancer care (interquartile range: 107900–308010 INR). The rates of catastrophic expenditure calculated using self-reported and estimated incomes were different (Fig 4). The rate of catastrophic expenditure for this cohort of patients was highest when the self-reported income was used (80.5%), when compared to the rates obtained with the IWI income (73.5%) or the Gapminder tool (56.7%).

## Discussion

The IWI and the Gapminder tool are feasible and promising instruments to assess household income in global health research studies. The vast majority of this cohort of colorectal patients

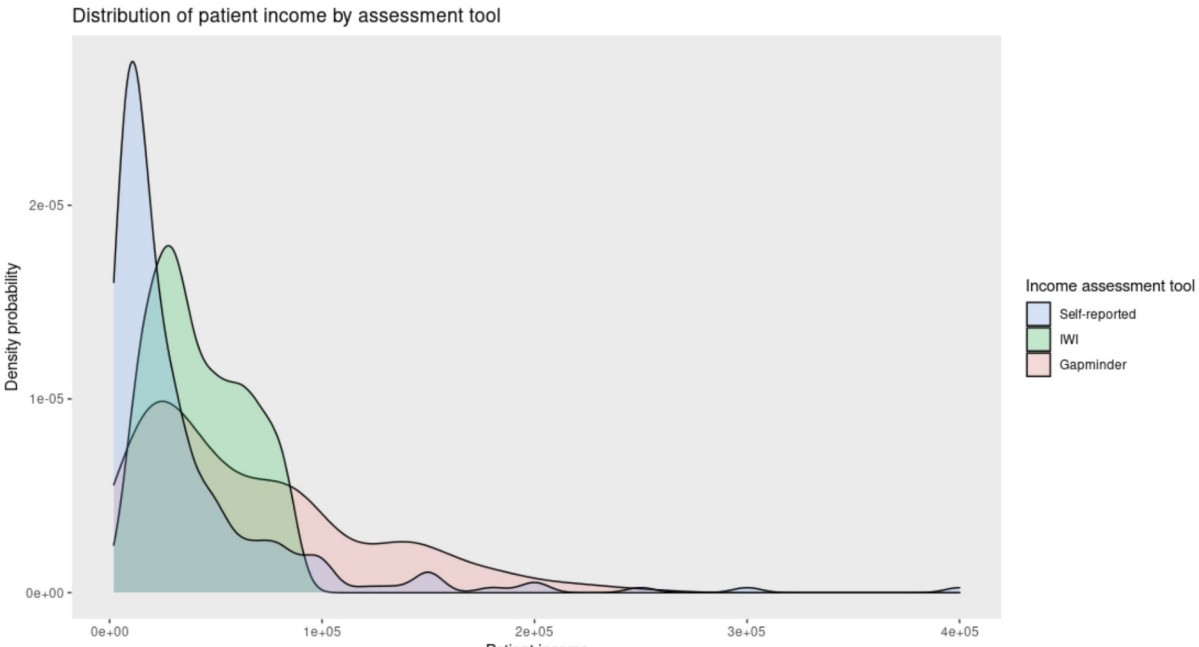

**Fig 2. Distribution of all patient household income assessments (self-reported, IWI and Gapminder) in a combined density plot.** Combined Kernel density plot graph displaying the distribution of incomes assessed by the three income assessment tools (self-reported, IWI and Gapminder tool). Patient income is reported in Indian Rupees (₹).

from India was able to complete both instruments. When using the IWI index to estimate household income, 92.8% of the patients will have an estimated income that differs up to 50000 INR from their self-reported income. The Gapminder tool shows worse agreement, with 88.9% of the patients having an estimated income that differs up to 100000 INR from the self-reported income. The use of different tools to assess income resulted in different catastrophic expenditure rates within the cohort, highlighting that further research and refinement of methods is needed to monitor this well-established Sustainable Development Goal at global scale.

The fact that self-reported incomes are generally lower than estimated incomes is unsurprising. Firstly, patients might feel uncomfortable revealing their income due to social pressures [8, 9, 11]. Secondly, all income data were collected at the point of inclusion in the cohort study which was, by definition, coincident with a new treatment decision for colorectal cancer. At this point in time, many patients could have been seeking funds from the hospital and the government to pay for their cancer care. Given that government and charity funds are usually allocated to patients based on their earnings, this might have played a role in patients underreporting their income [23]. Thirdly, the self-reported income was collected from household members wages and did not include any other sources of income (e.g. inheritance). Even though self-reported incomes tended to be lower, the most severe outlying income figures were actually self-reported. This can reflect the inability of the indirect tools to capture these incomes, but we can't exclude a previously described form of social desirability bias where some patients felt like they should overreport their incomes [11].

The International Wealth Index and the Gapminder tool were both designed to allow a comparison of wealth levels across households [17, 20]. It is expected that households generate wealth through their income but also through other sources of purchasing power. There is no

**(A)**

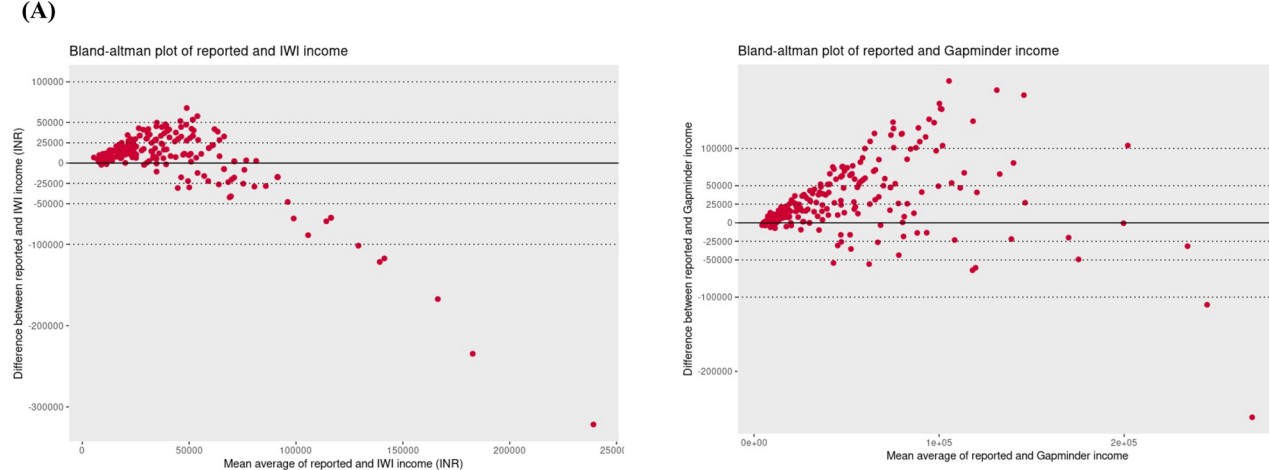

**(B)**

|  | Proportion of patients for whom the difference between the assessed incomes is | | |
|---|---|---|---|
|  | within a 25000 INR interval | within a 50000 INR interval | within a 100000 INR interval |
| **Reported and IWI income** | 67.1% (151/225) | 92.8% (209/225) | 97.3% (219/225) |
| **Reported and Gapminder income** | 49.3% (111/225) | 70.2% (158/225) | 88.9% (200/225) |

**Fig 3. Bland-Altman analysis exploring agreement between the self-reported income and the International Wealth Index and the Gapminder incomes. (A) Bland Altman plots describing the agreement patterns between the self-reported income and the International Wealth Index and the Gapminder incomes, respectively**. Plots display difference between measured incomes against mean average of measured incomes. The dotted lines represent a difference between income measurements of 25000, 50000 and 100000 INR, respectively. The full line represents perfect agreement (no difference between reported and estimated income). **(B) Proportion of patients for whom the observed difference between the assessed incomes is within 25000, 50000 and 100000 INR**.

consensus on the best definition and threshold for catastrophic expenditure and some authors argue that wealth and expenditure levels might be a better reflection of households' capacity to pay for cancer care [24]. The indirect tools have the advantage of assessing wealth more holistically, which might justify why IWI and Gapminder incomes being higher than the self-reported income. Additional factors can contribute to higher income figures with the IWI and the Gapminder tool, such as patients owning donated household goods that typically belong to higher income households or the personal choice to rent versus buy a house (renting requires less short-term investment for a better house).

The authors of the IWI have described that a truncation effect exists in this index, meaning that it has less discriminative power in very low and very high levels of wealth, as the score is limited to the number and value of the included assets [17]. This explains why the income figures obtained with the use of the IWI showed a narrower distribution. This can also explain why the agreement between the IWI and the self-reported income is worse for high income earners.

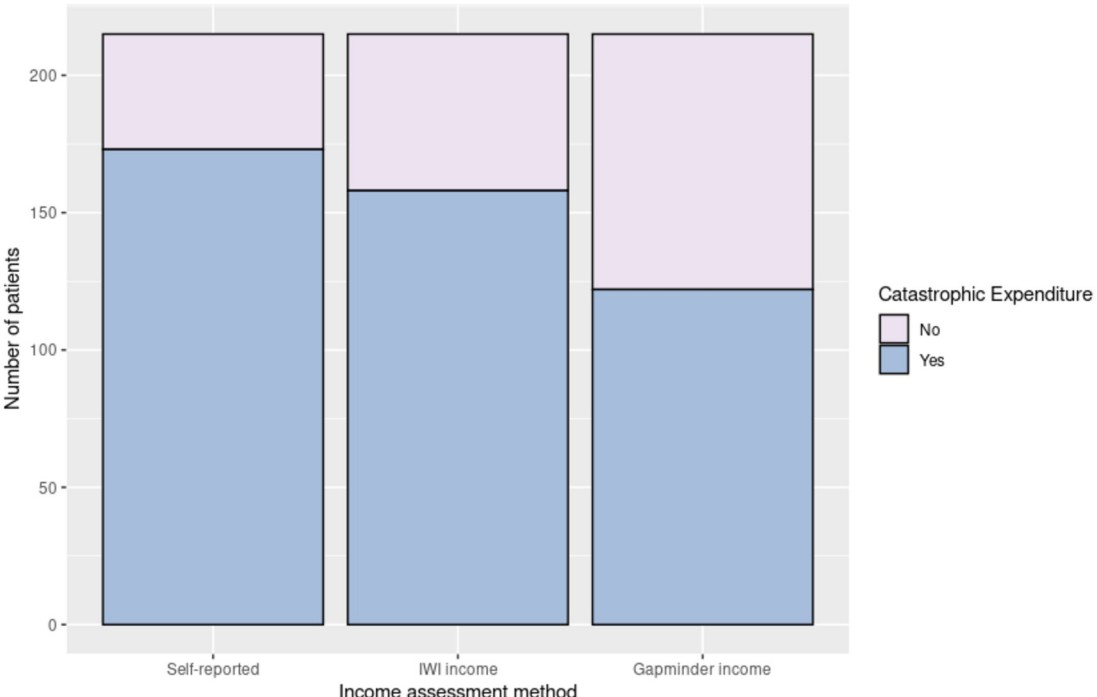

**Fig 4. Catastrophic expenditure rates for colorectal cancer care, when using self-reported household income versus the International Wealth Index income or the Gapminder income.** The bar chart compares the number of patients undergoing catastrophic expenditure, when the income is self-reported, assessed through the IWI or the Gapminder tool.

The Gapminder tool showed a less skewed distribution of incomes than self-reporting or the IWI, with more patients having higher incomes. This might reflect a better ability to discriminate wealth but can also be a result of this tool including multiple sources of data to match a household to an income (e.g. minimum and average wages, own food production, average consumption for specific groups of the population) [18]. The way that this tool was designed and used in the study could have contributed to worse agreement with self-reported income when compared to the IWI. Although there was a wide range of house items for the patient to choose from (around 30 items per category), patients could have struggled to find items that look like their own. We also acknowledge that similar pictures could be matched to quite different incomes, e.g. private flush toilets looking quite similarly were owned by households living on 1295 USD a month and 3477 USD a month, potentially resulting in very different income estimations.

Although this study is a unique evaluation of three different methods to assess household income to date, some limitations must be acknowledged. Firstly, although the completion rates were very high, full limitations of the use of the IWI and Gapminder tools were not explored in this study (e.g. biases and barriers to respond). Secondly, although this is a multi-centric cohort study, it only included tertiary care hospitals in India, limiting generalisability for other countries and other levels of care in India, in which prospective data would be ideal to assess their use. Finally, the IWI and the Gapminder tools revealed better agreement for low-income earning households, compromising the applicability of the results to high-income patients.

Future research should focus on the refinement of these tools, enabling their use worldwide. From our data, we can conclude that the IWI shows better agreement with self-reported

incomes, but high income earning households are underrepresented. The IWI was derived from a set of household surveys from which several items were selected and their association to household income was drawn into a weighted score. A refinement of this index to capture some items typically owned by wealthy households could improve its discriminative power for high-income earners and its overall generalisability. Regarding the Gapminder tool, the estimated incomes follow a distribution shape comparable to the self-reported income but are more dispersed. For future studies using this tool, a selection of fewer pictures for each category (e.g. ten kitchens representative of ten different income levels), representative of well-defined income levels can improve its ability to estimate accurate income figures.

A qualitative assessment of the use of these tools is necessary to achieve a better understanding of their strengths, limitations and potential biases. A consensus process involving patients, researchers and decision makers might be needed ahead of its global use, enabling the development of a tool that matches goals at individual and community levels.

Provided the necessary refinements, these tools have the potential to facilitate real-world assessment and monitoring of healthcare related catastrophic expenditure, without the need for direct income assessment. Moreover, data on house characteristics and household goods is routinely collected in several household surveys, meaning that the IWI and the Gapminder tool could allow large scale modelling of catastrophic expenditure rates from population survey data. Although more research is needed, the high completion rates and the apparent low complexity of both the IWI and the Gapminder tool allows us to conclude that they are promising tools in global health research.

## Conclusions

This study demonstrates that household income assessment can be performed with indirect tools, avoiding self-reported income. Although these alternative tools need refinement, they are promising assets for monitoring and modelling of catastrophic at global scale.

## Supporting information

**S1 Appendix. Details of the income assessment instruments and supplementary tables and figures.**
(DOCX)

## Author Contributions

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
