## [Decision Letter · Decision Letter 0]

22 Jun 2022

PONE-D-22-11886Feasibility of the International Wealth Index and the Gapminder tool as instruments to assess household income: a prospective patient-level cohort study in IndiaPLOS ONE

Dear Dr. Joana Filipa Ferreira Simoes,

Thank you for submitting your manuscript to PLOS ONE. After careful consideration, we feel that it has merit but does not fully meet PLOS ONE’s publication criteria as it currently stands. Therefore, we invite you to submit a revised version of the manuscript that addresses the points raised during the review process.

ACADEMIC EDITOR: 

I would like to congratulate the authors for approaching the catastrophic health expenditure estimations and income of households through novel tools available. It is a known fact that income based qusetionnaire items are predominantly over/under reported. Therefore, additional tools utility in assessment is a welcome step. However, to proceed further within the context of the scientific validity of the manuscript, please do rebutt/rexplain/ review  the various points as raised by the reviewers. 

Specifically ,the issue as raised by reviewer 2 about PPP and conversions (already discussed in paragraph 1 page no 6) needed to be re explained / re discussed & with clarity, so that it is easily comprehensible at the reader level also. Morever, please re check the referencing style as mentioned by reviewer 1 as per the PLOS ONE Journal style apart from other suggestions.

Editor Specific: it is gently proposed that manuscript , "The overall goal of this cohort study

was to determine catastrophic expenditure and treatment adherence rates among

colorectal cancer patients in India" , was mentioned in the material and methods section. But, there was no mention/discussion of the 'adherence rates' as such in the manuscript. Please re consider about mentioning of the same in the methods. Moreover, if possible please try to include CHE in the title also. 

We look forward to receiving your revised manuscript.

Kind regards,

Gopal Ashish Sharma, MBBS, MD

Academic Editor

PLOS ONE

Journal Requirements:

"None."

5. One of the noted authors is a group or consortium CROCODILE study group. In addition to naming the author group, please list the individual authors and affiliations within this group in the acknowledgments section of your manuscript. Please also indicate clearly a lead author for this group along with a contact email address.

Reviewers' comments:

Reviewer's Responses to Questions

**Comments to the Author**

1. Is the manuscript technically sound, and do the data support the conclusions?

Reviewer #1: Yes

Reviewer #2: Partly

2. Has the statistical analysis been performed appropriately and rigorously? 

Reviewer #1: Yes

Reviewer #2: Yes

3. Have the authors made all data underlying the findings in their manuscript fully available?

Reviewer #1: Yes

Reviewer #2: No

4. Is the manuscript presented in an intelligible fashion and written in standard English?

Reviewer #1: Yes

Reviewer #2: Yes

5. Review Comments to the Author

Reviewer #1: The topic is really relevant and addressing income is very sensitive issue, therefore indirect assessment is needed still few things are required to be addressed for betterment of the article as suggested in the review.

Reviewer #2: The paper on the feasibility of the IWI and the Gapminder Tool to assess household income is well-written and makes an interesting argument in favor of using wealth indexes rather than self-reported income. I have a few concerns about the findings:

1. It seems that the self-reported income is reported in rupees while the conversions of the wealth index is reported in PPP. It is not clear that these values are comparable.

2. It is not clear to me how the equations were obtained that converted the wealth Index and Gapminder index to income in PPP. It would be good if the authors explained how they derived these equations in more detail and the justification for using these.

3. The authors should present a review of the literature and summarize whether other articles have made comparisons between self-reported income and wealth indexes.

6. PLOS authors have the option to publish the peer review history of their article (what does this mean?). If published, this will include your full peer review and any attached files.

Reviewer #1: **Yes: **Dr. Satabdi Mitra

Reviewer #2: No

---

## [Author Response · Author response to Decision Letter 0]

22 Aug 2022

Reviewer #1: The topic is really relevant and addressing income is very sensitive issue, therefore indirect assessment is needed still few things are required to be addressed for betterment of the article as suggested in the review.

Thank you very much. We have addressed the reviewers’ comments in the attached manuscript:

1: People with long-standing illness suffer from catastrophic expenditure, direct and indirect of which cancer plays a major part.

A consideration of non-communicable diseases and cancer patients being at high risk of catastrophic expenditure was included in the introduction: “Worldwide, more than 900 million people are experiencing healthcare related catastrophic expenditure and households with non-communicable diseases, particularly cancer are at high risk” line (71-72)

2: Rationality of the sample size needs to be clarified.

The sample size calculations were clarified according to the protocol: “The sample size was defined at the time of protocol design, assuming an anticipated proportion of cancer patients suffering catastrophic expenditure around 45% [13,14]. For a prespecified absolute precision of 10% and 5% (error margins recommended by the United Nations for household surveys), a respective sample size of 95– 380 patients would be required at a confidence interval of 95% [15]. A pragmatic trade-off was decided according to usual recruitment numbers at colorectal cancer multidisciplinary teams in the included hospitals, allowing for a feasible study length and clinical relevance.” line 129-138

3: The study findings need to be repeated in the discussion section.

We have summarised the study findings in the first paragraph of the discussion, while trying to avoid repetition: “The IWI and the Gapminder tool are feasible and promising instruments to assess household income in global health research studies. The vast majority of this cohort of colorectal patients from India was able to complete both instruments. When using the IWI index to estimate household income, 92.8% of the patients will have an estimated income that differs up to 50000 INR from their self-reported income. The Gapminder tool shows worse agreement, with 88.9% of the patients having an estimated income that differs up to 100000 INR from the self-reported income. The use of different tools to assess income resulted in different catastrophic expenditure rates within the cohort, highlighting that further research and refinement of methods is needed to monitor this well-established Sustainable Development Goal at global scale”. Line 324-334. We are happy to be guided by the reviewers and the editorial team if further changes are desirable.

4: Usefulness of conducting a prospective study needs to be emphasised. 

We have now emphasised the usefulness of prospective data in the discussion: “Secondly, although this is a multicentric cohort study, it only included tertiary care hospitals in India, limiting generalisability for other countries and other levels of care in India, in which prospective data would be ideal to assess their use.” line 389-391

5: After the authors’ name, a full stop should be given. If number authors more than 6, et al to be mentioned. 

The references are now corrected according to the journal style.

Reviewer #2: The paper on the feasibility of the IWI and the Gapminder Tool to assess household income is well-written and makes an interesting argument in favour of using wealth indexes rather than self-reported income. I have a few concerns about the findings:

1. It seems that the self-reported income is reported in rupees while the conversions of the wealth index is reported in PPP. It is not clear that these values are comparable.

Thank you for your comment. The monetary conversion of the international wealth index was drawn from a formula that was derived in united states dollars (USD). The PPP were used to exchange USD into indian rupees (INR), to make the figures comparable with reported incomes (reported by patients in INR). We have clarified this in the methods “The formula above correlates the IWI with an income figure in 2015 United States Dollars (USD). In order to obtain a figure that is comparable to the income reported by patients in our study in 2021 (Indian Rupees), the IWI income in USD was multiplied by purchase power parities (PPP) for USD-INR exchange and by the consumer price index to account for inflation between 2015 and 2021.” line 191-195

2. It is not clear to me how the equations were obtained that converted the wealth Index and Gapminder index to income in PPP. It would be good if the authors explained how they derived these equations in more detail and the justification for using these.

Thank you very much. The formula describing the relationship between each one of the IWI items and household income was published by the Gapminder foundation, where several ways of assessing household income are explored in depth. We have clarified the details of the formula derivation in the paper: “This formula was obtained by computing the IWI in multiple population based household surveys (including the India Human Development Survey but also population datasets from other low-middle income countries), on the basis of the available information about households and assets. The formula estimates how an increase in the IWI score is associated with an expected average increase in household income.” line 182-187

The Gapminder Foundation provides a figure (in USD) for household income for each picture of house, kitchen, toilet, floor and bed and therefore a translation into income wasn’t necessary. Their methods for assessing household income are described on their website and were bespoke to the interviewees, including reported income, reported consumption, owned assets, minimum and average wages for people’s occupations and benefits available through social welfare. The only formula we used was to get the mean average of the incomes matched to each picture selected by the patient, with PPP to allow exchange into INR. We have clarified this in the methods: “The methods used by the Gapminder team to assess household income and match them to the pictures were varied, including: reported income, reported consumption, owned assets, minimum and average wages for people’s occupations and benefits available through social welfare.” line 200-204

3. The authors should present a review of the literature and summarize whether other articles have made comparisons between self-reported income and wealth indexes.

Thank you very much, this was included in the introduction now: ‘A search on PubMed and Google Scholar was performed identifying one study where an asset index was assessed as a tool to predict income quintiles with low performance in African populations and an exploratory analysis from data from Mexico showing that concluding that assets might better reflect wealth as they fluctuate less over time[12, 13]. A systematic review exploring the relationship between wealth indexes and socio-economic status demonstrated that they are likely to be distinct measures[14]. No previous studies were found, in which asset or house characteristics were used to estimate household income figures for catastrophic expenditure calculations.” line 98-106

---

## [Decision Letter · Decision Letter 1]

5 Oct 2022

Feasibility of the International Wealth Index and the Gapminder tool as instruments to assess household income and estimate catastrophic expenditure: a prospective patient-level cohort study in India

PONE-D-22-11886R1

Dear Dr. Joana Filipa Ferreira Simoes,

We’re pleased to inform you that your manuscript has been judged scientifically suitable for publication and will be formally accepted for publication once it meets all outstanding technical requirements.

Kind regards,

Gopal Ashish Sharma, MBBS, MD

Academic Editor

PLOS ONE

Additional Editor Comments (optional):

Considering the scientific suitability as reveiwed , the current manuscript would further stimulate /facilitate similar research to the context. 

Reviewers' comments:

Reviewer's Responses to Questions

**Comments to the Author**

1. If the authors have adequately addressed your comments raised in a previous round of review and you feel that this manuscript is now acceptable for publication, you may indicate that here to bypass the “Comments to the Author” section, enter your conflict of interest statement in the “Confidential to Editor” section, and submit your "Accept" recommendation.

Reviewer #1: All comments have been addressed

Reviewer #2: All comments have been addressed

2. Is the manuscript technically sound, and do the data support the conclusions?

Reviewer #1: Yes

Reviewer #2: Yes

3. Has the statistical analysis been performed appropriately and rigorously? 

Reviewer #1: Yes

Reviewer #2: Yes

4. Have the authors made all data underlying the findings in their manuscript fully available?

Reviewer #1: Yes

Reviewer #2: Yes

5. Is the manuscript presented in an intelligible fashion and written in standard English?

Reviewer #1: Yes

Reviewer #2: Yes

6. Review Comments to the Author

Reviewer #1: The work is very apt and addressed an important issue. All suggestions have been addressed nicely and included in revised version.

Reviewer #2: All comments have been addressed.

7. PLOS authors have the option to publish the peer review history of their article (what does this mean?). If published, this will include your full peer review and any attached files.

Reviewer #1: **Yes: **Dr. Satabdi Mitra

Reviewer #2: No

---

## [Editor Report · Acceptance letter]

21 Nov 2022

PONE-D-22-11886R1 

Feasibility of the International Wealth Index and the Gapminder tool as instruments to assess household income and estimate catastrophic expenditure: a prospective patient-level cohort study in India 

Dear Dr. Simoes:

I'm pleased to inform you that your manuscript has been deemed suitable for publication in PLOS ONE. Congratulations! Your manuscript is now with our production department. 

Kind regards, 

on behalf of

Dr. Gopal Ashish Sharma 

Academic Editor

PLOS ONE